# Ciliary chemosensitivity is enhanced by cilium geometry and motility

David Hickey[1], Andrej Vilfan[1,2]*, Ramin Golestanian[1,3]*

[1]Max Planck Institute for Dynamics and Self-Organization (MPIDS), Göttingen, Germany; [2]J. Stefan Institute, Ljubljana, Slovenia; [3]Rudolf Peierls Centre for Theoretical Physics, University of Oxford, Oxford, United Kingdom

**Abstract** Cilia are hairlike organelles involved in both sensory functions and motility. We discuss the question of whether the location of chemical receptors on cilia provides an advantage in terms of sensitivity and whether motile sensory cilia have a further advantage. Using a simple advection-diffusion model, we compute the capture rates of diffusive molecules on a cilium. Because of its geometry, a non-motile cilium in a quiescent fluid has a capture rate equivalent to a circular absorbing region with ~4× its surface area. When the cilium is exposed to an external shear flow, the equivalent surface area increases to ~6×. Alternatively, if the cilium beats in a non-reciprocal way in an otherwise quiescent fluid, its capture rate increases with the beating frequency to the power of 1/3. Altogether, our results show that the protruding geometry of a cilium could be one of the reasons why so many receptors are located on cilia. They also point to the advantage of combining motility with chemical reception.

## Introduction

Cilia are small hairlike organelles with a microtubule-based core structure that protrude from the cell surface. They are found on most eukaryotic cells (*Nachury and Mick, 2019*) and can be broadly classified into two categories: primary and motile. Primary cilia, of which there is only one on each cell, have primarily sensory functions (as receptors for chemical, mechanical, or other signals) (*Zimmermann, 1898*; *Berbari et al., 2009*; *Hilgendorf et al., 2016*; *Spasic and Jacobs, 2017*; *Ferreira et al., 2019*). Due to their shape and their role in signalling, they are often referred to as 'the cell's antenna' (*Marshall and Nonaka, 2006*; *Malicki and Johnson, 2017*). Motile cilia, typically appearing in larger numbers (*Brooks and Wallingford, 2014*; *Spassky and Meunier, 2017*), move the surrounding fluid by beating in an asymmetric fashion (*Golestanian et al., 2011*; *Gilpin et al., 2020*), and often with some degree of coordination (*Uchida and Golestanian, 2010*; *Elgeti and Gompper, 2013*). They play a key role in a number of processes, including the swimming and feeding of microorganisms (*Guasto et al., 2012*; *Lisicki et al., 2019*), mucus clearance in airways (*Bustamante-Marin and Ostrowski, 2017*), fluid transport in brain ventricles (*Faubel et al., 2016*), and egg transport in Fallopian tubes. However, there are exceptions to this classification. Primary cilia in the vertebrate left-right organiser are motile and drive a lateral fluid flow that triggers, through a mechanism that is not yet fully understood, a distinct signalling cascade determining the body laterality (*Essner et al., 2002*). There is also mounting evidence that motile cilia can have various sensory roles (*Bloodgood, 2010*), including chemical reception (*Shah et al., 2009*). Adversely, receptors localised on motile cilia, such as ACE2, can also act as entry points for viruses including SARS-CoV-2 (*Lee et al., 2020*). Some chemosensory systems, including vomeronasal (*Leinders-Zufall et al., 2000*) and olfactory neurons (*Bhandawat et al., 2010*) and marine sperm cells (*Kaupp et al., 2003*), are known to achieve a sensitivity high enough to detect a small number of molecules.

The sensitivity of a chemoreceptor is characterised by its binding affinity for the ligand, as well as its association/dissociation kinetics. If the time-scale of ligand dissociation is longer than the time-

*For correspondence:
andrej.vilfan@ds.mpg.de (AV);
Ramin.Golestanian@ds.mpg.de (RG)

Competing interests: The authors declare that no competing interests exist.

scale of the changes in ligand concentration, or if the ligands bind irreversibly, the sensitivity is determined by the binding rate alone. It has been shown that the theoretical limit of sensing accuracy is achieved when the receptors detect the frequency of binding events and when re-binding is excluded (*Bialek and Setayeshgar, 2005*; *Endres and Wingreen, 2009*) Because diffusion is fast on very short length scales, only 1% of the surface area of a cell or cilium needs to be covered in high-affinity receptors to obtain near-perfect adsorption (*Berg and Purcell, 1977*). Even if this condition is not satisfied, the membrane itself could non-specifically bind the ligands with near-perfect efficacy, which then reach the receptors in a two-stage process. In either of these cases, as long as there is no advection, the binding rates can be estimated using the theory of diffusion-limited reactions (*Adam and Delbruck, 1968*). This binding rate is known as the diffusion limit, and it has already been shown that flagella-driven swimming microorganisms can break the diffusion limit in order to increase their access to nutrients (*Short et al., 2006*).

The increasingly overlapping functions of sensory and motile cilia lead to the natural question about the advantage of placing receptors on a cilium, or in particular on a motile cilium. Because of its small volume, a cilium forms a compartment that facilitates efficient accumulation of second messengers (*Marshall and Nonaka, 2006*; *Hilgendorf et al., 2016*). Placing receptors on a protrusion, away from the flat surface, could have other advantages, like avoiding the effect of surface charges or the glycocalyx. It has also been suggested that the location of chemoreceptors on cilia exposes them to fluid that is better mixed (*Marshall and Nonaka, 2006*). A recent study suggests that the hydrodynamic interaction between motile and sensory cilia can enhance the sensitivity of the latter (*Reiten et al., 2017*). However, the question of how the geometry and motility of cilia affect their ability to capture and detect ligands has still remained largely unexplored.

In this paper, we investigate the theoretical limits on association rates of ligands on passive and motile cilia. In particular, we address the question of whether the elongated shape of a cilium and its motility can improve its chemosensory effectiveness. By using analytical arguments and numerical simulations, we show that the capture rate of a cilium is significantly higher than that of a receptor located on a flat epithelial surface. Motile cilia can further improve their chemosensitivity. Finally, we show that a cilium within an immotile bundle has a lower capture rate than an isolated cilium, but a higher one when the cilia are sufficiently motile.

## Results

In this study, we calculate the second-order rate constant for diffusive particle capture on a cilium. We discuss scenarios where the fluid and the cilium are at rest, where the fluid exhibits a shear flow, where the cilium is actively beating, and where a bundle of hydrodynamically interacting cilia absorbs particles.

We consider a perfectly absorbing cilium protruding from a non-absorbing surface, in a fluid containing some chemical species with a concentration field $c$. Far from the cilium, the unperturbed concentration has a constant value $c_0$. The rate constant $k$ is defined such that

$$I = c_0 k, \tag{1}$$

where $I$ is the capture rate, defined as the number of captured particles per unit time.

Since the aforementioned cilium is perfectly absorbing, we define an absorbing boundary condition such that the concentration of the chemical species is zero at every point on the cilium's surface. We assume that the flat membrane surrounding the cilium does not absorb particles and it is therefore described with a reflecting boundary condition at $z = 0$. The geometry and boundary conditions are illustrated in *Figure 1*.

### Cilium in quiescent fluid

We consider a cilium (modelled as a cylinder next to a boundary at $z = 0$) in a quiescent fluid, with the goal of determining its capture rate constant in the absence of advection. In the case where there is a steady state with no advection, the advection-diffusion equation reduces to

$$D\nabla^2 c = 0, \tag{2}$$

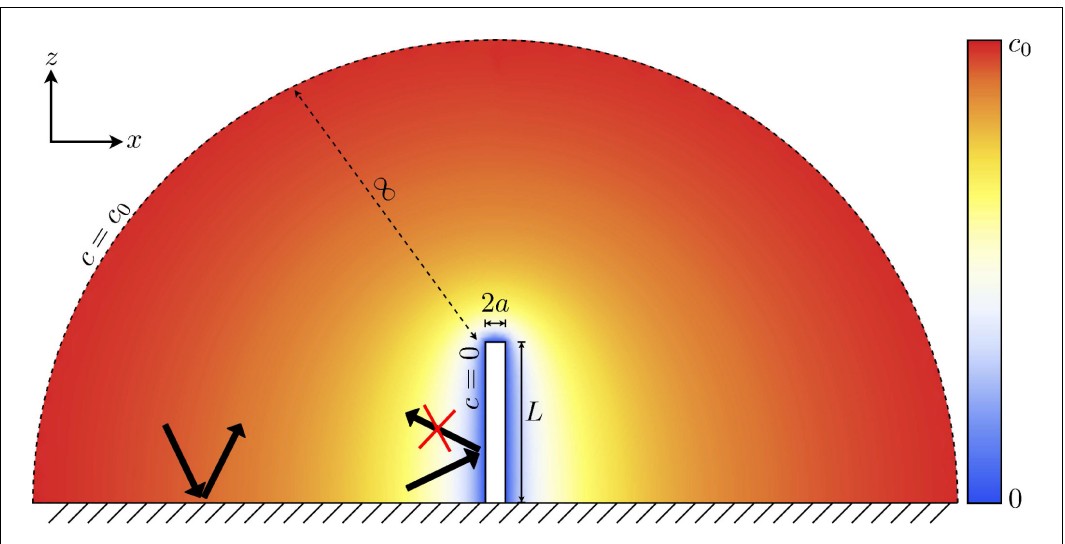

**Figure 1.** The concentration boundary conditions and general setup of the problem to be solved. The cilium satisfies an absorbing boundary condition, and there is a constant concentration an infinite distance from the cilium. The coloured overlay shows the concentration field in the absence of any fluid flow.

where $D$ is the diffusion constant. The rate constant is determined by the integral of the current density $\mathbf{J}$ over the surface, which follows from Fick's law:

$$k = -\frac{1}{c_0}\int d\mathbf{S}\cdot\mathbf{J} = \frac{1}{c_0}\int d\mathbf{S}\cdot(D\nabla c). \tag{3}$$

As we show in Appendix 1, the rate constant can be evaluated using an analogy between particle diffusion and electrostatics (*Berg and Purcell, 1977*). Up to a prefactor, the capture rate is determined by the self-capacitance $C$ of a conducting body of the same shape as $k = DC/\varepsilon_0$.

To determine the capture rate of a cilium embedded in a non-absorbing surface, we first eliminate the reflective boundary condition at the surface by symmetrically extending the problem to a cylinder of length $2L$ in open space and considering $1/2$ of its capacitance. There is no closed-form expression for the capacitance of a cylinder, so we loosely approximate this cylinder as a prolate spheroid with semi-major axis $L$ and semi-minor axis $a$. Using its self-capacitance in the limit $L \gg a$ (*Snow, 1954*), we find the rate constant:

$$k_{\text{cilium}} = 2\pi D \frac{L}{\ln(2L/a)}. \tag{4}$$

This value agrees well with simulations: the ratio of the simulated to this analytical rate constant is 1.02.

The finding that the capture rate scales almost linearly with the length of the cilium can be compared to experimental data obtained on olfactory cilia from the nasal cavity of mouse, whose lengths in different regions vary from a few micrometers to tens of micrometers. *Challis et al., 2015* have used patch-clamp recordings on olfactory sensory neurons and measured the response to pulses of an odorant (eugenol or a mixture of 10 odorants), lasting $5-400\,\text{ms}$. Regions with different lengths show very different sensitivity thresholds, differing by an order of magnitude. The results are qualitatively consistent with the predicted length dependence of the capture rate.

To quantify the advantage of localising the receptors on a cilium, we compare it with a case where the receptors form a circular patch on a flat surface (*Figure 2a*). Again, we assume that the receptor patch has a perfectly absorbing surface, while the surface surrounding it is reflective. We determine the size of the patch needed to attain the same rate constant as the cilium. The rate constant for a circular patch on the reflective boundary can be found by applying the electrostatic analogy to the well-known result for the self-capacitance of a thin conducting disc of radius $R$ (*Berg and Purcell, 1977*):

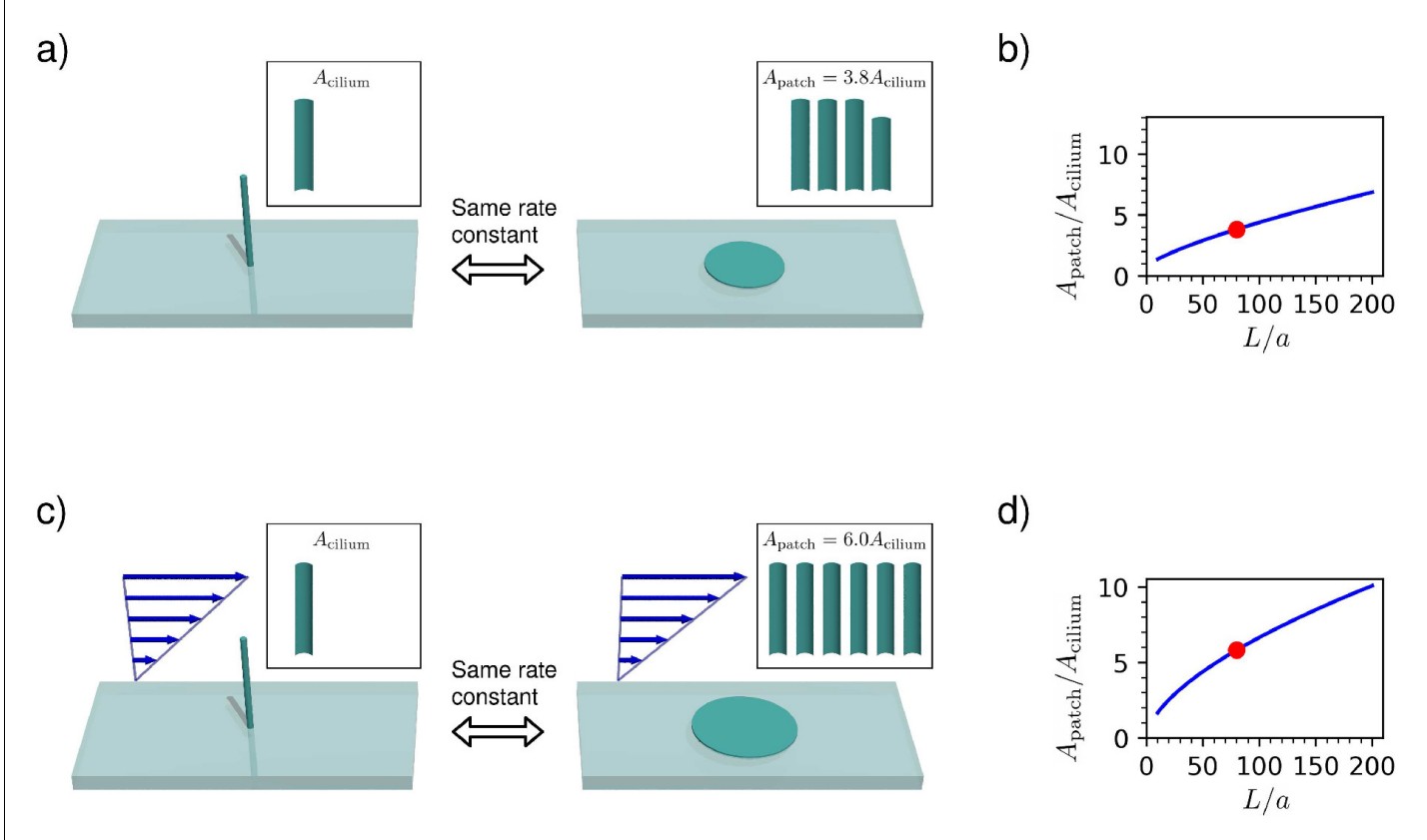

**Figure 2.** Comparison between capture rates of a non-motile cilium and a circular patch on the surface. All diagrams use $L/a = 80$, indicated on the graphs by a red dot. (a) In a quiescent fluid, the cilium has the same capture rate as a surface patch with 3.8 times the surface area. (b) The area ratio $A_{\mathrm{patch}}/A_{\mathrm{cilium}}$ as a function of the cilium aspect ratio $L/a$ in a quiescent fluid, given by *Equation (6)*. (c) In a shear flow at a high Péclet number, the capture rate of the cilium reaches that of a surface patch with 6.0 times the surface area. (d) The area ratio as a function of the aspect ratio in the high Péclet number limit (*Equation 14*).

The online version of this article includes the following source data and figure supplement(s) for figure 2:

**Figure supplement 1.** Capture rate as a function of the Péclet number for passive cilia in a shear flow, obtained from numerical simulations.

**Figure supplement 1—source data 1.** Event counts and calculated rates as shown in *Figure 2—figure supplement 1*.

$$k_{\mathrm{patch}} \approx 4DR. \tag{5}$$

We find that the patch has a much larger surface area than the cilium with the same rate constant. We can calculate this area ratio:

$$\frac{A_{\mathrm{patch}}}{A_{\mathrm{cilium}}} \approx \frac{\pi^2}{8} \cdot \frac{L}{a} \frac{1}{\ln^2(2L/a)}. \tag{6}$$

The area ratio as a function of the aspect ratio $L/a$ is shown in *Figure 2b*. For a typical cilium aspect ratio of $L/a = 80$ (with $L = 10\,\mu\mathrm{m}$, and $a = 125\,\mathrm{nm}$), this area ratio is 3.8, implying that the cilium is much more effective per unit area than a receptor on the surface of the cell. Olfactory cilia have a great variation of lengths, ranging from $2.5\,\mu\mathrm{m}$ to $100\,\mu\mathrm{m}$ (*Challis et al., 2015*; *Williams et al., 2014*). If we neglect the fact that long cilia are not straight, the calculated area ratio ranges from 2.7 to 18. Using an exact numerical result for the capacitance of a cylinder (*Paffuti, 2018*), the ratio becomes 4.5 for $L/a = 80$. With the dimensions given above, the radius of the circular patch with the same capture rate is $R = 3.4\,\mu\mathrm{m}$.

## Cilium in shear flow

At the scale of cilia, the flow is characterised by a low Reynolds number, meaning that viscous forces dominate over inertia. The fluid motion is well-described by the Stokes equation, together with the incompressibility condition:

$$\eta \nabla^2 \mathbf{u} - \nabla p = 0 \tag{7}$$

$$\nabla \cdot \mathbf{u} = 0 \tag{8}$$

in which $\mathbf{u}$ is the fluid velocity, $\eta$ is the dynamic viscosity, and $p$ is the pressure. The concentration field of some chemical species suspended within this fluid is governed by the advection-diffusion equation:

$$\frac{\partial c}{\partial t} + \mathbf{u} \cdot \nabla c = D \nabla^2 c \tag{9}$$

where $c$ is a function of both position and time.

The ratio of advection to diffusion is described by the dimensionless Péclet number. This is usually written as some characteristic flow speed multiplied by some characteristic length scale, all divided by the diffusion constant.

Because the cilium grows from a flat surface with a no-slip boundary condition, the flow can be described as a uniform shear flow with the shear rate $\dot{\gamma}$. To estimate the capture rate constant of a cilium in a shear flow, we make use of the fact that the radius of the cylinder is much smaller than the length scale over which the shear flow varies. We therefore approximate the local rate density at any point on the cilium with that of an infinitely long cylinder in a uniform flow with velocity $v(z) = \dot{\gamma}z$. The capture rate per unit length is

$$\frac{\mathrm{d}k_{\text{cilium}}}{\mathrm{d}z} = \beta D \cdot \left( A \frac{av}{D} \right)^{1/3}, \tag{10}$$

where $\beta = 2.50$ is a numerical constant (see Appendix 2 for derivation).

Now the total rate constant is obtained by integration over the cilium length

$$k_{\text{cilium}} = \int_0^L \mathrm{d}z \, \beta D \cdot \left( \frac{a\dot{\gamma}z}{D} \right)^{1/3} = \frac{3}{4} \beta DL \cdot \left( \frac{L}{a} \right)^{-1/3} \text{Pe}_{\text{cilium}}^{1/3}. \tag{11}$$

We take the characteristic velocity to be the speed of the cilium's tip relative to the surrounding fluid, and hence the Péclet number for the extended cilium is

$$\text{Pe}_{\text{cilium}} = \frac{\dot{\gamma}L^2}{D}. \tag{12}$$

This expression for the rate once again shows a strong positive relationship between cilium length and sensitivity, as is known to be the case in real biological systems (*Challis et al., 2015*). The characteristic Péclet number for the cross-over between the diffusive and the convective capture is of the order $\sim L/a \approx 80$.

Once again we determine the size of a circular surface patch offering an equivalent effectiveness to the cilium in a flow with the same shear rate (*Figure 2c*). The high-Pe rate constant for a patch in a shear flow is (*Stone, 1989*):

$$k_{\text{patch}} \approx DR \left[ \zeta \cdot \text{Pe}_{\text{patch}}^{1/3} + O\left( \text{Pe}_{\text{patch}}^{-1/6} \right) \right], \tag{13}$$

where $\text{Pe}_{\text{patch}} \equiv \dot{\gamma}R^2/D$ and $\zeta = 2.157$ is a purely numerical constant. We can calculate the ratio of the area of the equivalent patch to the area of the cilium for these high-Pe asymptotic results:

$$\frac{A_{\text{patch}}}{A_{\text{cilium}}} \approx \frac{1}{2} \left( \frac{3\beta}{4\zeta} \right)^{6/5} \cdot \left( \frac{L}{a} \right)^{3/5} \approx 0.42 \left( \frac{L}{a} \right)^{3/5}. \tag{14}$$

The area ratio is shown in *Figure 2d* and compared with the results in a quiescent fluid. For a

typical cilium aspect ratio $L/a = 80$ (with $L = 10\,\mu\mathrm{m}$, and $a = 125\,\mathrm{nm}$), this area ratio is 6.0 – much larger than the area ratio in a quiescent fluid, which was 3.8. This means that a cilium is better per area than a patch at both low and high Péclet numbers, but the cilium excels when the Péclet number is large.

We additionally investigated the question how robust the results are if the receptors are localised to only one segment of the cilium. In a model, we assumed that of the total length $L$, the distal part $\nu L$ is absorbing, while the proximal $(1 - \nu)L$ is reflective. At high Péclet numbers, we can modify the integration limits in *Equation (11)* and obtain a theoretical capture rate $k_{\mathrm{cilium}}(\nu) = k_{\mathrm{cilium}} \cdot (1 - (1 - \nu)^{4/3})$. The result is compared to simulations in *Figure 2—figure supplement 1*. A cilium with receptors over the distal 50% of its length therefore achieves 60% of the maximal capture rate.

## Active pumping

A mounting collection of evidence suggests that both primary and motile cilia have sensory roles (*Bloodgood, 2010*). We are interested in the extent to which cilium motility can increase their ability to detect particles. To this end, we numerically simulate various different possible types of ciliary motion in otherwise quiescent fluids.

Because of the complex flow patterns and time-dependent boundary conditions, the absorption by a beating cilium is not analytically tractable. Instead, we use numerical simulations to determine the rate constants. We consider four different active pumping scenarios: a purely reciprocally moving cilium, a cilium tracing out a cone around an axis perpendicular to the surface, a cilium tracing out a tilted cone, and a cilium with a trajectory that includes bending, to raise the pumping efficiency (all shown in their respective order in *Figure 3a–d*).

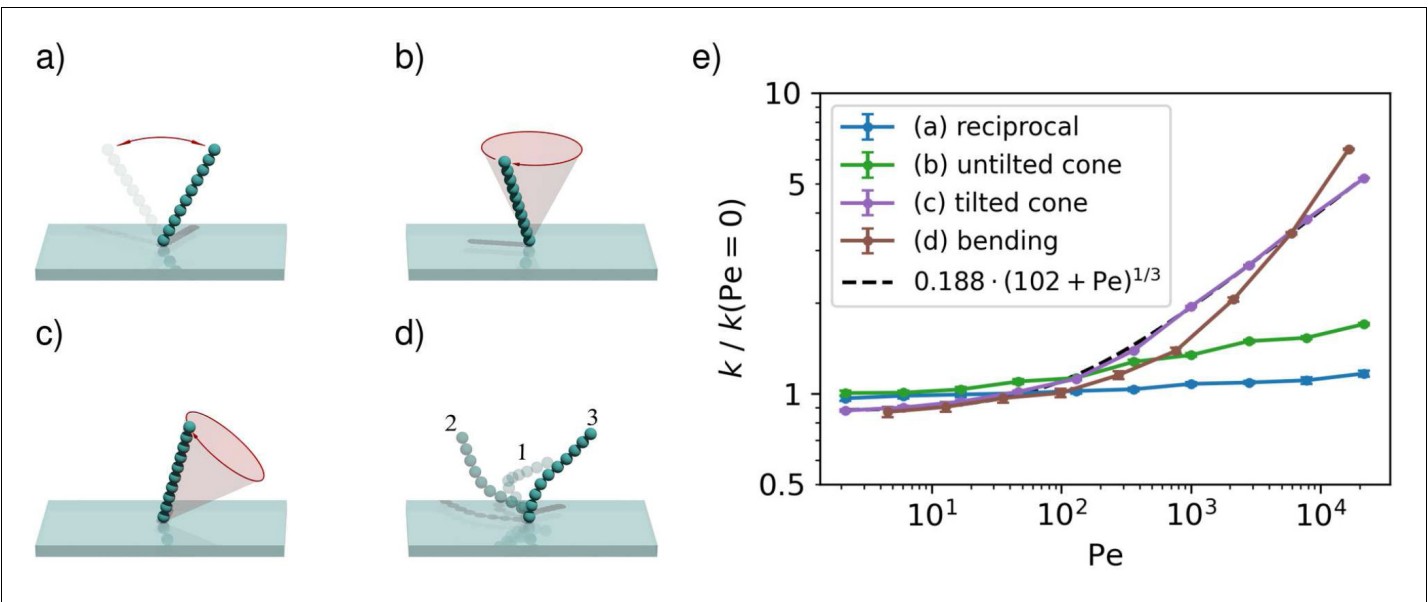

**Figure 3.** The capture rate of an active cilium for four types of motion. (**a**) The cilium is undergoing reciprocal motion, which is not generating any net flow. (**b**) The cilium moves along a cone with its axis perpendicular to the surface, such that it produces a rotational flow, but no long-range fluid transport. (**c**) The cilium moves along a tilted cone, which generates a long-range volume flow. (**d**) The cilium follows a realistic trajectory, beginning with a recovery stroke along the no-slip surface in 1, then performing an overhead power-stroke from 2 to 3 before returning to one in another recovery stroke. (**e**) The capture rate constants $k$ of a beating cilium as a function of the Péclet number. The rates are determined using stochastic simulations. The error bars denote 95% confidence intervals and the dashed line shows a fit function that interpolates between the high and low-Péclet limits. All rates are normalised to the rate constant for a diffusion-limited capture with a cylindrical cilium with the same length and width.

The online version of this article includes the following source data and figure supplement(s) for figure 3:

**Source data 1.** Event counts and calculated rates as shown in *Figure 3e*.

**Figure supplement 1.** Capture rate of an actively beating cilium tracing out a tilted cone, plotted as a function of the Péclet number.

**Figure supplement 1—source data 1.** Event counts and calculated rates as shown in *Figure 3—figure supplement 1*.

The rate constants in these scenarios, relative to that of a non-moving cilium, are plotted in *Figure 3e*. Analogously to the cilium in a shear flow, we define the Péclet number using the maximum tip velocity during the cycle:

$$\mathrm{Pe} = \frac{v_{\mathrm{tip}}^{\max} L}{D}. \qquad (15)$$

The reciprocally moving cilium (*Figure 3a*) displays almost no improvement over several orders of magnitude of the Péclet number. This is expected, because Purcell's scallop theorem (*Purcell, 1977*) states that purely reciprocal motion does not create any net flow, so the particle intake is largely diffusive in nature. A minor increase of the rate constant with the Péclet number is caused by the local shear flow that facilitates absorption on the surface.

The cilium moving around a vertical cone (*Figure 3b*) induces a net rotational flow, but no inflow or outflow (by symmetry, the time-averaged flow can only have a rotational component [*Vilfan, 2012*]). Nevertheless, the constant motion of the cilium through the fluid leads to a higher local capture efficiency. The rate constant therefore shows more improvement; over a few orders of magnitude of the Péclet number, the rate constant increases by a factor of two.

The tilted cone (*Figure 3c*) shows a much higher capture rate, which is unsurprising. When the cilium is near to the plane, the no-slip boundary screens the flow, whereas when it is far from the plane, its pumping is unimpeded. This results in the cilium inducing a long range flow in one direction, characterised by a finite volume flow rate (*Smith et al., 2008*). The long range flow causes a constant intake that replenishes the depleted particles. At high Péclet numbers, the capture rate scales $k \sim \mathrm{Pe}^{1/3}$, which is the same dependence as in an external shear-flow, although with a prefactor that is lower by a factor of ~2. Locally, the relative flow around the cilium is the same whether a cilium is pivoting or resting in a shear flow. The pumping effect of the tilted cilium, on the other hand, provides sufficient inflow that the concentration around a cilium sees only a small depletion effect.

We finally simulated the capture process on a cilium exerting a realistic beating pattern, consisting of a stretched working stroke and a bent, sweeping recovery stroke (*Figure 3d*). The capture rate is close to that of the tilted cone, but surpasses it at very high Péclet numbers.

## Collective active pumping

We consider seven cilia on a hexagonal centred lattice with lattice constant $0.95L$, with a view to understand how the presence of multiple cilia affects performance. We quantify the performance gain using a quantity $Q$, which we define as

$$Q = \frac{k_{\mathrm{multiple}}}{k_{\mathrm{cilium}}(\mathrm{Pe}) \cdot N_{\mathrm{cilia}}}, \qquad (16)$$

which represents the fractional per-cilium improvement in rate constant compared to a single isolated cilium at the same Péclet number.

Using numerical simulations, we find that at zero Péclet number (*Figure 4a–b*), $Q \approx 0.5$, which means that the cilia locally deplete the concentration field, harming the per-cilium effectiveness; in a quiescent fluid, it is most efficient for cilia to stand far away from their neighbours.

However, when the cilia actively move (with each tracing out a tilted cone with a different randomly-chosen phase lag compared to its neighbours, as in *Figure 4e*) the trend is reversed: we find that at $\mathrm{Pe} \approx 10000$, $Q \approx 1.53$, meaning that per cilium, the capture rate is around 50% higher in the collective when compared to an isolated cilium with the same Péclet number. We find that over the range of Péclet numbers simulated, the $Q$ increases monotonically with the Péclet number (*Figure 4h*).

When comparing these randomly chosen phases to a patch of cilia which beat in uniform, we find that cilia which beat in phase (*Figure 4d*) see an improvement over the stationary case with $Q \approx 1.16$, but are much less effective than the cilia patch that beats with random phases. The random phases give a higher volume flow, and complex hydrodynamic interactions between the randomly-phased cilia result in a slightly higher capture chance for any given particle. Similar levels of improvement are also seen for other arrangements of cilia forming a bundle, that is, $N_{\mathrm{cilia}} = 19$ on a hexagon (*Figure 4f*) or $N_{\mathrm{cilia}} = 4$ on a square (*Figure 4g*). Furthermore, an improvement of randomly beating over uniformly-beating cilia has been observed in previous work with finite-sized particles (*Ding and*

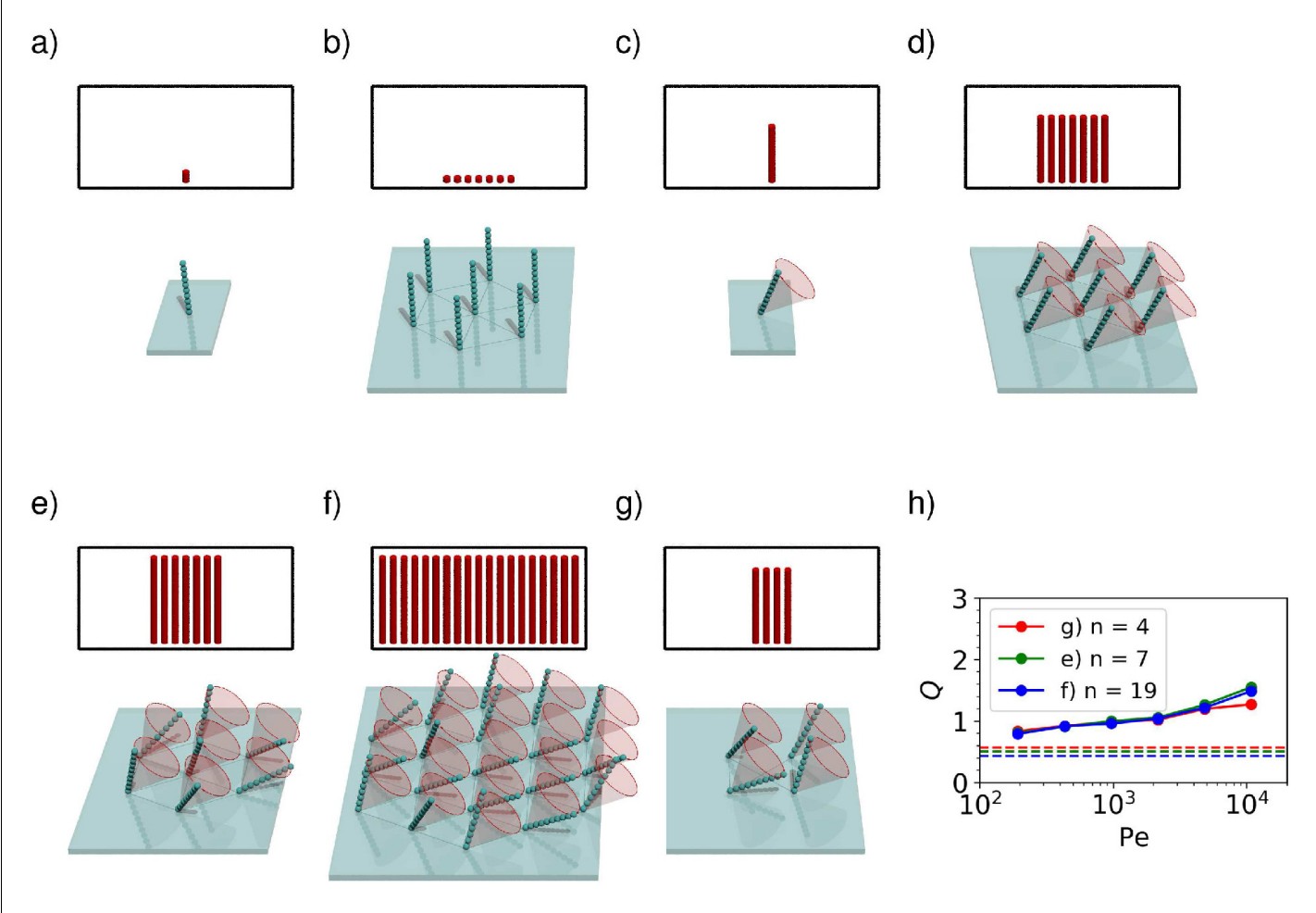

**Figure 4.** Comparison between the capture rate constant of a single cilium (**a, c**) and a bundle of $N_{cilia} \in \{4, 7, 19\}$ cilia (**b, d–g**). In the insets, the height of each red cylinder indicates the rate constant per cilium at $Pe \approx 10000$, and the number of cylinders represents the number of cilia. For immotile cilia (**a, b**), a bundle has a lower per-cilium capture rate than an isolated cilium, although the the total rate constant of the bundle is higher. The reduced capture rate per cilium is caused by the depletion of ligands close to the bundle. For motile cilia (**d–g**), the situation is reversed and the capture rate per cilium in a bundle (**d–g**) can be significantly higher than for an isolated cilium (**c**). The increase can be explained by the collective flow generation, which helps the capture on all cilia. In (**d**) the cilia all beat with the same frequency corresponding to $Pe \approx 10000$ but with identical phases. In (**e–g**) all cilia beat with the same frequency corresponding to $Pe \approx 10000$, but their phases are chosen randomly. It can be seen that the random phases give a higher rate constant than the uniform phases. (**h**) shows how the performance gain $Q$ varies with the Péclet number for different configurations. The rates shown at each point are the average of 30 random phase configurations like the one shown in (**e**). The dashed line is the $Q$-value for $Pe = 0$ for each configuration.

The online version of this article includes the following source data for figure 4:

**Source data 1.** Event counts and calculated rates as shown in *Figure 4h*.

*Kanso, 2015*; *Nawroth et al., 2017*). The results suggest that mutual enhancement of capture rates is a robust phenomenon and does not depend on a specific geometry.

## Discussion

Our results address a simple question: does the location of so many chemical receptors on cilia bring them an advantage in sensitivity? Besides the well-known advantages of compartmentalisation, which facilitates the downstream signal processing, we show that the elongated shape of a cilium provides an advantage for the capture rate of molecules in the surrounding fluid. The advantages can be summarised as follows:

1. If neither the fluid nor the cilium move and the process of particle capture is purely diffusive, the elongated shape improves the capture rate of the cilium by giving it better access to the diffusing ligands. The length dependence of the capture rate has the sub-linear form $k \sim L/\log L$. With typical parameters, the cilium achieves a capture rate equivalent to that of a circular patch of receptors on a flat surface with $4\times$ the surface area of the cilium.

2. When a non-moving cilium is exposed to a shear flow, the advantage increases, mainly because the tip of the cilium is exposed to higher flow velocities. The capture rate scales with $k \sim L^{4/3}$ and becomes equivalent to that of a surface patch with approximately $6\times$ the surface area at high flow rates.

3. An actively beating cilium can achieve capture rates comparable to those by a passive cilium in a shear flow with the same relative tip velocity, but only if the beating is non-reciprocal, that is, if the cilium generates a long range directed flow. The capture rate can scale with the beating frequency to the power of 1/3 or higher.

4. Without motility, a bundle of sensory cilia achieves a capture rate *per cilium* that is lower than that of a single cilium, because of the locally depleted ligand concentration. However, the situation can become reversed if the cilia are beating: then each cilium benefits from the flow produced by the bundle as a whole, and the per-cilium capture rate can be significantly higher than in an isolated beating cilium. Cilia beating with random phases achieve significantly higher capture rates than when beating in synchrony.

Our results are based on a few assumptions. We assumed that the particles get absorbed and detected upon their first encounter of the cilia surface – an assumption that is justified if the receptors are covering the surface at a sufficient density (*Berg and Purcell, 1977*), or if the particles bind non-specifically to the membrane of the cilium first. We also treat the particles as point-like (their size only has an influence on their diffusivity), which is accurate for molecules up to the sizes of a protein and we do not expect a significant error even for small vesicles. The Rotne-Prager tensor approximation used to determine the flow fields does not exactly satisfy the no-slip boundary condition on the surface of the cilium, especially at high Péclet numbers.

With the typical dimensions of a cilium ($L = 10\,\mu\mathrm{m}$, $a = 0.125\,\mu\mathrm{m}$) and a diffusion constant of a small molecule $D = 10^{-9}\,\mathrm{m^2s^{-1}}$, we obtain $k_{\mathrm{cilium}} = 7\,\mathrm{pM^{-1}s^{-1}}$. A chemosensory cilium working at the physical limit is therefore capable of detecting picomolar ligand concentrations on a timescale of seconds. Sensitivity thresholds in the sub-picomolar range have been measured in some olfactory neurons (*Frings and Lindemann, 1990*; *Zhang et al., 2013*), indicating that some olfactory cilia work close to the theoretical sensitivity limit. If the cilia are embedded in mucus with a viscosity at least 3 orders of magnitude higher than water (*Lai et al., 2009*) (we disregard its viscoelastic nature here) and the molecule has a Stokes radius of a few nanometres, the diffusion-limited capture rate reduces to around $k_{\mathrm{cilium}} = 1\,\mathrm{nM^{-1}s^{-1}}$.

In a shear flow with a typical shear rate of $\dot{\gamma} = 10\,\mathrm{s^{-1}}$, the Péclet number of a small molecule in water is of the order of $\mathrm{Pe} \approx 1$, where the capture rate still corresponds to the stationary case. However, with larger molecules and higher viscosities, the Péclet numbers can exceed $10^4$, leading to a significant enhancement of the capture rate.

When the same cilium is beating with a frequency of $25\,\mathrm{Hz}$, the Péclet number is of the order $\sim 10$, which is too small to have an effect on the capture rate. With larger molecules and higher viscosities, the Péclet numbers can be significantly higher. With a medium viscosity of $0.2\,\mathrm{Pa\cdot s}$ (200 times the water viscosity) and a Stokes radius of $10\,\mathrm{nm}$, it reaches $10^5$, meaning that the motility accelerates the capture rate by one order of magnitude. For example, according to one hypothesis, motile cilia in the zebrafish left-right organizer (Kupffer's vesicle) both generate flow and detect signalling particles, possibly extracellular vesicles (*Ferreira et al., 2017*; *Ferreira et al., 2019*) similar to the proposed 'nodal vesicular parcels' (*Tanaka et al., 2005*). With a cilium length of $L = 6\,\mu\mathrm{m}$ and a particle radius of $a = 100\,\mathrm{nm}$, we obtain $\mathrm{Pe} = 1300$, showing that the capture rates can be several times higher than in a passive cilium. *Figure 5* shows how the molecular Stokes radius affects the fluid viscosity required to break the diffusion limit for a few different scenarios. However, when the particle size becomes comparable to the cilium diameter, the approximation that treats them as point particles loses validity. Indeed, it has been shown that particle size can have a direct steric effect on the capture rate (*Ding and Kanso, 2015*). Furthermore, the capture process of large particles can depend on a competition between hydrodynamic and adhesive forces (*Tripathi et al., 2013*). Steric effects can even lead to particle enrichment in flow compartments (*Nawroth et al., 2017*).

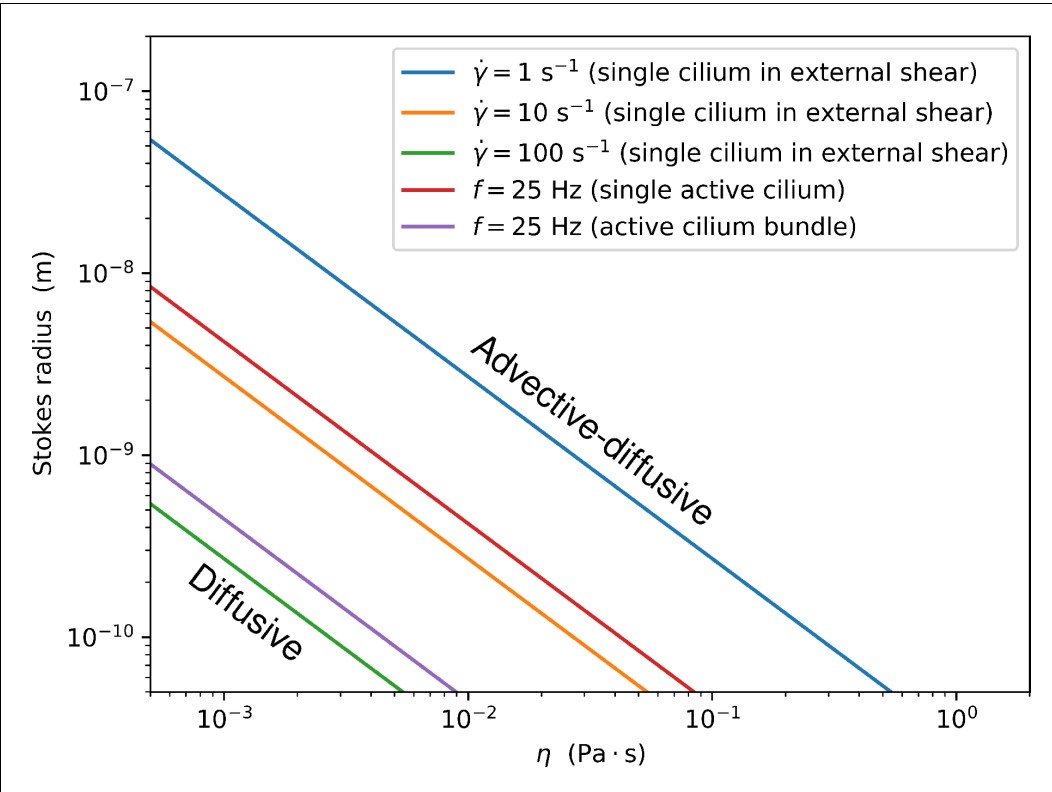

**Figure 5.** The demarcation between the regime where the rate constant is determined mostly by the diffusion limit and the regime in which it is enhanced by advection as a function of the fluid viscosity η and the particle Stokes radius. The blue, orange, and green lines show the results for a passive cilium in a shear flow (*Figure 2c*), the red line for an actively beatig cilium (*Figure 3c*) and the magenta line for a bundle of 7 cilia (*Figure 4e*). For all lines, the cilium dimensions are $L = 10\,\mu$m and $a = 250\,$nm.

We have thus proven that for individual isolated cilia the geometry of a cilium always means an advantage in chemical sensitivity over receptors covering the same area on a flat surface (assuming they act as perfect absorbers), whether in a quiescent or moving fluid. At high Péclet numbers, which are achieved in viscous fluids, with very large particles or in very strong flows, the advantage of a cilium increases further and even confers an advantage in chemosensitivity to cilium bundles over individual cilia. These advantages can work in concert with others, such as avoiding charged surfaces and glycocalix and the provision of a closed compartment on the inside. Further work might examine the extent to which motility benefits cilia in a fluid with bulk flow, or investigate the effect of metachronal waves on ciliary chemosensitivity. Finally, our results shed light on possible engineering applications for microfluidic sensing devices based on these ideas, for example using magnetic actuation (*Vilfan et al., 2010*; *Meng et al., 2019*; *Matsunaga et al., 2019*).

## Materials and methods

Numerically simulated point particles are injected into a finite system containing a motile cilium, and move around due to advection (resulting from the motion of the cilium) and diffusion, until they either escape from the system or are absorbed by the cilium. The proportion of particles which are captured is used to compute a rate constant.

### Flow calculation

The hydrodynamics are computed using a modified Rotne-Prager mobility tensor $\mathbf{M}$ that accounts for the no-slip boundary. If there are $N$ spheres of equal radius $R$ in the simulation, each having a

prescribed trajectory $\mathbf{r}_i(t)$ and each acted upon by a force $\mathbf{F}_i(t)$, then these forces must satisfy (*Vilfan et al., 2010*)

$$\dot{\mathbf{r}}_i(t) = \sum_{j=1}^{N} \mathbf{M}\big[\mathbf{r}_i(t), \mathbf{r}_j(t); R, R\big] \cdot \mathbf{F}_j(t) \tag{17}$$

for every $i \in [1, N]$. Since every term except the forces is known, the forces can be determined numerically at a given $t$ by solving this set of simultaneous equations. Then the fluid velocity at any point $\mathbf{x}$ can be determined by

$$\mathbf{u}(\mathbf{x}, t) = \sum_{i=1}^{N} \mathbf{M}[\mathbf{x}, \mathbf{r}_i(t); 0, R] \cdot \mathbf{F}_i(t). \tag{18}$$

In the simulations we used $N = 20$ spheres, corresponding to an aspect ratio $L/a = 40$. A somewhat lower value than in the analytical calculations was chosen to save computational time and also to compensate for the fact that a cylinder is replaced with a chain of spheres.

## Injection

We require a particle injection procedure that satisfies the concentration boundary condition $c \rightarrow c_0$ far from the absorbing cilium. We achieve this by introducing two bounding boxes in the simulation: an inner and an outer box, separated by a thin distance $d$ (*Figure 6*). The particles are injected at the boundary of the inner box and absorbed at the outer box. The injection rate is calculated such that it corresponds to the advective-diffusive flux through the layer between the boxes if the concentration at the inner box is $c_0$. Because the flux through the boundary layer is much larger than the flux of particles absorbed inside the inner box, the method is suited to ensure a constant concentration boundary condition. The method is similar to a recent algorithm using a single boundary (*Ramírez-Piscina, 2018*), but uses a simpler injection function.

To calculate the injection current density, we solve the one-dimensional steady-state advection-diffusion equation

$$0 = D\frac{\mathrm{d}^2 c}{\mathrm{d}x^2} - v\frac{\mathrm{d}c}{\mathrm{d}x}, \tag{19}$$

with the boundary conditions $c(0) = 0$ and $c(d) = c_0$. The solution is

$$c(x) = c_0 \frac{e^{vxD} - 1}{e^{vdD} - 1}. \tag{20}$$

By the application of Fick's law, this leads to an expression for the current density through the inner box:

$$j(x) = vc_0 \frac{1}{1 - e^{-vdD}}. \tag{21}$$

We assume that a test particle will take take much longer to reach the cilium than the characteristic time required for the flow to change, and hence we take $v = \langle \mathbf{u}(\mathbf{x}, t) \cdot \hat{\mathbf{n}} \rangle_t$, where $\hat{\mathbf{n}}$ is the inward pointing surface normal of the inner box. This function can then be used to probabilistically weight where particles are injected on the inner box.

## Numerical integration

The test particle position is updated using an Adams-Bashforth-Milstein multistep numerical integration method in the presence of noise (*Tocino and Senosiain, 2015*):

$$\mathbf{x}_{i+1} = \mathbf{x}_i + \Delta t \left[\frac{3}{2}\mathbf{u}(\mathbf{x}_i, t) - \frac{1}{2}\mathbf{u}(\mathbf{x}_{i-1}, t - \Delta t)\right] + \xi_i. \tag{22}$$

Because the computation of the flow field $\mathbf{u}$ (see *Equation 18*) is the most demanding step, it is advantageous over methods that require additional function evaluations per step. $\xi_i$ is a vector

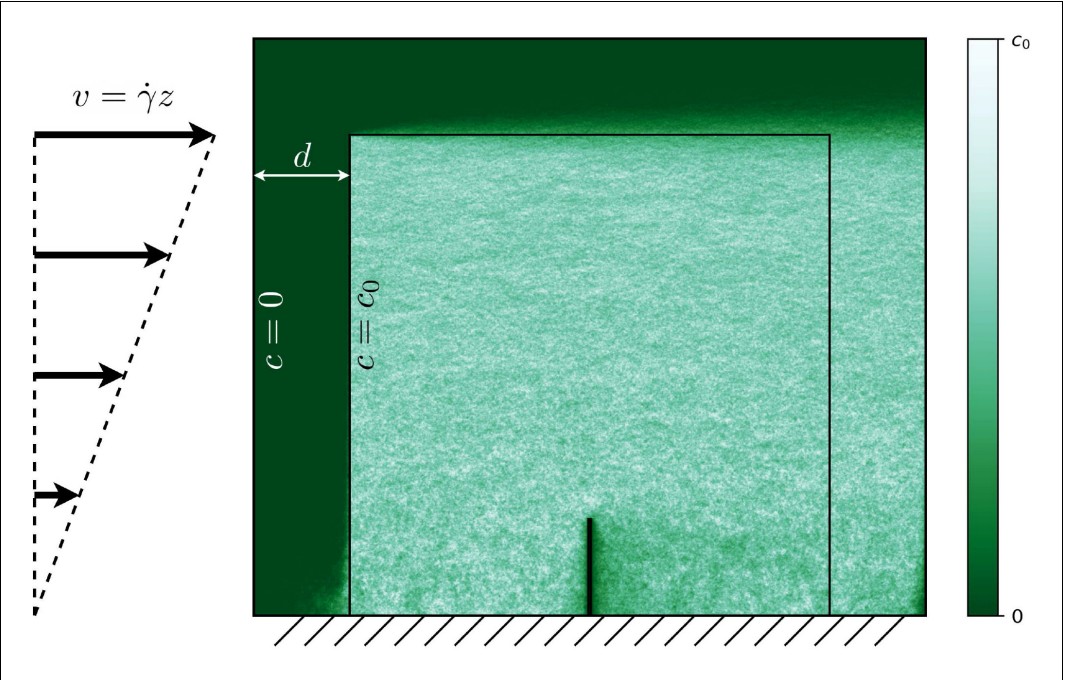

**Figure 6.** The boundary conditions used for injection. Since the fraction in incident particles absorbed by the cilium is small compared to the fraction absorbed by the outer surface, the concentration at the inner boundary is very close to $c_0$. The coloured overlay shows the concentration as recorded in an example numerical simulation of a cilium in a shear flow with $\mathrm{Pe} = 50$.

where each element is pseudorandomly generated Gaussian noise with standard deviation $\sqrt{2D\Delta t}$ and mean of zero.

### Rate evaluation

We finish each simulation run when the particle position reaches the cilium (capture), or the outer box (escape). At the end, the rate constant is determined as

$$k = \frac{I}{c_0} \frac{n_{\text{capture}}}{n_{\text{capture}} + n_{\text{escape}}}, \tag{23}$$

where $I$ is the calculated total particle flux, obtained by integrating the flux density over the inner box, $I = \int j \, \mathrm{d}S$.

### Numerical parameters

For all numerical simulations, we use a cilium consisting of 20 beads (thus giving a length to radius ratio $L/a = 40$). For the conical and reciprocal motion (*Figure 3a–c*), we use an opening angle (between the cone axis and surface) of 30°, and for the tilted conical motion (*Figure 3c*) the axis of the cone is tilted relative to the vertical by an angle of 55°.

In the collective regime, the parameters are the same, with the addition of a hexagon lattice constant of $0.95L$. The cones are tilted such that their axes are perpendicular to one chosen side of the hexagon (left to right in *Figure 4d-f*).

## Acknowledgements

We thank David Zwicker for comments on the manuscript. This work has been supported by the Max Planck Society. A.V. acknowledges support from the Slovenian Research Agency (grant no. P1-0099).

## Additional information

### Funding

| Funder | Grant reference number | Author |
|---|---|---|
| Max-Planck-Gesellschaft | | David Hickey<br>Andrej Vilfan<br>Ramin Golestanian |
| Javna Agencija za Raziskovalno Dejavnost RS | P1-0099 | Andrej Vilfan |

The funders had no role in study design, data collection and interpretation, or the decision to submit the work for publication.

### Author contributions

David Hickey, Data curation, Software, Formal analysis, Validation, Investigation, Visualization, Methodology, Writing - original draft, Writing - review and editing; Andrej Vilfan, Conceptualization, Software, Formal analysis, Supervision, Validation, Visualization, Methodology, Writing - original draft, Writing - review and editing; Ramin Golestanian, Conceptualization, Supervision, Funding acquisition, Methodology, Project administration, Writing - review and editing

### Author ORCIDs

David Hickey (iD) https://orcid.org/0000-0003-0149-712X
Andrej Vilfan (iD) https://orcid.org/0000-0001-8985-6072
Ramin Golestanian (iD) https://orcid.org/0000-0002-3149-4002

### Decision letter and Author response

Decision letter https://doi.org/10.7554/eLife.66322.sa1
Author response https://doi.org/10.7554/eLife.66322.sa2

## Additional files

### Supplementary files

• Transparent reporting form

### Data availability

All data generated or analysed during this study are included in the manuscript and supporting files. Source data files have been provided for Figures 3 and 4.

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

## Appendix 1

### Electrostatic analogy for the capture rate

In the following we explain the analogy between the capture rate of a diffusive particles and the self-capacitance in electrostatics (*Berg and Purcell, 1977*). The diffusion equation reads

$$D\nabla^2 c = 0,\tag{24}$$

where $D$ is the diffusion constant and $c$ the particle concentration. The boundary conditions are $c = c_0$ at infinity and $c = 0$ at the particle surface. The diffusion equation is equivalent to the Laplace equation for source-free electrostatics, in which the electrostatic potential $\phi$ obeys

$$\nabla^2 \phi = 0.\tag{25}$$

If the surface of the body in the electrostatic case has a potential of $-V_0$, the boundary conditions are equivalent as well. The rate constant is determined by the integral of the current density $\mathbf{J}$ over the surface, which follows from Fick's law:

$$k = -\frac{1}{c_0}\int d\mathbf{S}\cdot\mathbf{J} = \frac{1}{c_0}\int d\mathbf{S}\cdot(D\nabla c).\tag{26}$$

In the electrostatic version of the problem, the equivalent expression for self-capacitance is

$$C = \frac{q}{V_0} = \frac{1}{V_0}\int d\mathbf{S}\cdot(\varepsilon_0\nabla\phi),\tag{27}$$

where $-q$ is the charge on the body. By analogy, the rate constant can be expressed as:

$$k = \frac{D}{\varepsilon_0}C.\tag{28}$$

The electrostatic equivalence allows us to translate the calculation of the capture rates to a capacitance problem with a greater number of available solutions in the literature.

## Appendix 2

### Capture rate of a cylinder in flow

In the following, we calculate the capture rate of a cylinder, moving transversely through the flow at a high Péclet number. While defining the problem, one encounters the Stokes paradox, namely that the lateral mobility of an infinite cylinder at zero Reynolds number diverges. We avoid the problem by calculating the capture rate for a prescribed force density on the cylinder, which gives a well-defined near-field flow. Later, we can use the well established resistive force theory to estimate the force density at a given local velocity. A further simplification we make is to swap the boundary conditions (*Masoud and Stone, 2019*), such that the cylinder emits particles, leading to a concentration $c_0$ at its surface and 0 in the incoming fluid. We describe the flow around the cylinder with radius $a$ with the following stream function in cylindrical coordinates

$$\psi = -\frac{af}{8\pi\eta}\left(\frac{r}{a} - \frac{a}{r} - \frac{2r}{a}\ln\left[\frac{r}{a}\right]\right)\sin(\theta),\tag{29}$$

where $f$ is the force per unit length. The unperturbed fluid is coming from the $\theta = 0$ direction (*Appendix 2—figure 1*). The fluid velocity is determined as the curl of the stream function, e.g.:

$$v_\theta = \frac{\partial\psi}{\partial r}.\tag{30}$$

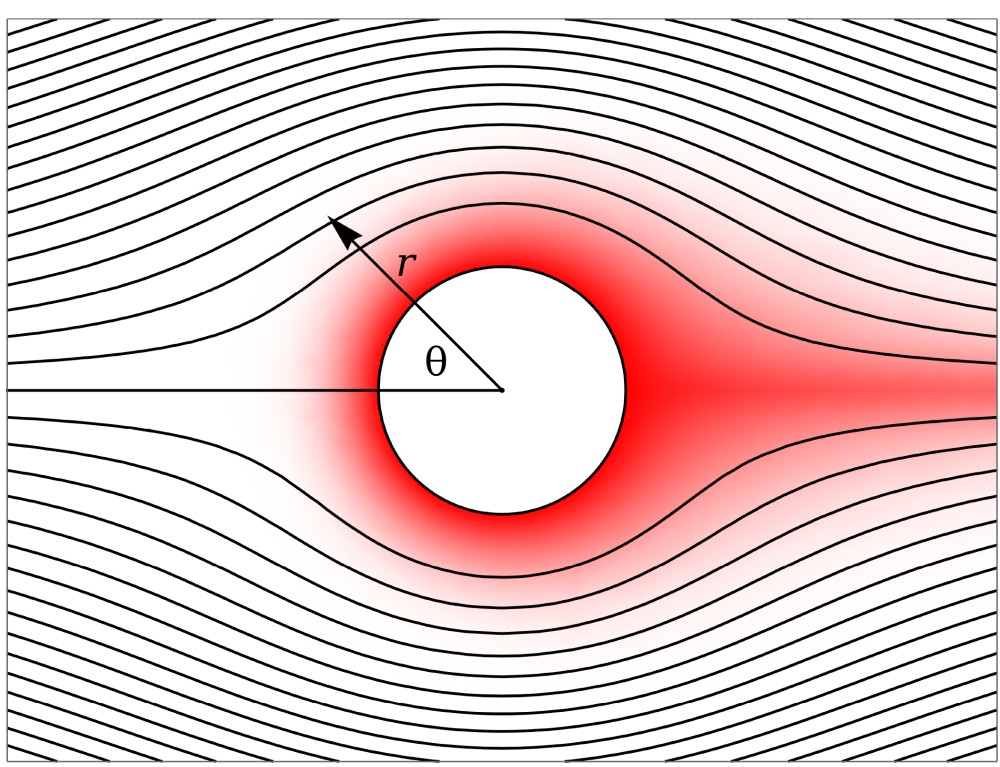

**Appendix 2—figure 1.** Streamlines (lines with a constant value of the stream function $\psi$) of the flow around a cylinder (black) and the concentration $c$ of emitted particles (red).

In the limit of a high Péclet number, the emitted particles stay in a thin boundary layer around the cylinder before escaping at $\theta = \pi$. We can therefore use the following approximation that only takes into account the leading order contribution to the stream function

$$\psi = \frac{2af}{8\pi\eta}\left(\frac{r-a}{a}\right)^2\sin(\theta).\tag{31}$$

In the following, we derive a partial differential equation for the particle flux $\Phi(\theta)$ across a radial half-plane starting with radius $r$ at the angle $\theta$, defined as

$$\Phi(r,\theta) = \int_r^\infty \mathrm{d}r'\, v_\theta(r')c(r') = \int_{\psi(r)}^\infty \mathrm{d}\psi\, c(\psi). \tag{32}$$

At a high Péclet number, advection dominates over diffusion, which only needs to be considered in the direction perpendicular to the stream lines, but not along. Due to flux conservation, the variation of $\Phi$ with the angle $\theta$ is caused by the diffusive transverse flux, driven by the concentration gradient

$$\left.\frac{\partial\Phi}{\partial\theta}\right|_\psi = -rD\frac{\partial c}{\partial r}. \tag{33}$$

We finally arrive at the PDE for the particle flux

$$\frac{\partial\Phi}{\partial\theta} = -rD\frac{\partial\psi}{\partial r}\frac{\partial^2\Phi}{\partial\psi^2} = A\sqrt{\psi}\sqrt{\sin\theta}\cdot\frac{\partial^2\Phi}{\partial\psi^2} \tag{34}$$

with the constant $A = 8D\sqrt{\pi\eta/af}$. The boundary conditions are $\Phi = 0$ for $\theta = 0$, reflecting zero flux at the inflow, while the fixed concentration at the surface, $c_0 = 1$, implies $\partial\Phi/\partial\psi|_{\psi=0} = -1$. A transformation of variables $t = \int \mathrm{d}\theta A\sqrt{\sin\theta}$ with $t(0) = 0$ and $t(\pi) = \sqrt{8/\pi}\,\Gamma[3/4]^2$ leads to

$$\frac{\partial\Phi}{\partial t} = \sqrt{\psi}\frac{\partial^2\Phi}{\partial\psi^2}, \tag{35}$$

which has the solution

$$\Phi(t,\psi) = t^{2/3}\bar{\Phi}\left(\frac{\psi}{t^{2/3}}\right) \tag{36}$$

with

$$\bar{\Phi}(x) = -\frac{x\Gamma\left[-\frac{2}{3},\frac{4}{9}x^{3/2}\right]}{\Gamma\left[-\frac{2}{3}\right]} \quad\text{and}\quad \bar{\Phi}(0) = \left(\frac{3}{2}\right)^{4/3}\left(\Gamma\left[\frac{1}{3}\right]\right)^{-1}. \tag{37}$$

An example of a particle concentration $c$ resulting from this solution is shown in **Appendix 2—figure 1**. The emission rate (equivalent to capture rate) per unit length is given by twice the particle flux (for two sides of the cylinder):

$$\frac{dk}{dz} = 2\Phi(t(\pi),0) = 3\left(\frac{6}{\pi}\right)^{1/3}\frac{\Gamma(3/4)^{4/3}}{\Gamma(1/3)}D\,\mathrm{Pe}_f^{1/3} = 1.822D\,\mathrm{Pe}_f^{1/3} \quad\text{with}\quad \mathrm{Pe}_f = \left(\frac{2af}{\pi\eta D}\right)^{1/3} \tag{38}$$

A previous calculation that used a similar approach, but solved the PDE with an approximate function, rather than the exact solution derived here, gave the prefactor 1.63 when converted to our units (**Friedlander, 1957**).

Finally, we can use the resistive force theory to estimate the force density per unit length as $f = C_N v \approx 1.3\pi\eta v$ and arrive at **Equation (10)** in the main text. The prefactor depends on the width-to-length ratio of the object and we used a value that gives a good result for typical ciliary dimensions (**Vilfan, 2012**).

