## [Decision Letter]

**Acceptance summary:**

The cilium is a conserved organelle found in most cell types, broadly classifiable as motile or primary cilia. For a long time, the function of the latter had been thought to be vestigial, until recent studies have revealed their importance for signalling. The authors suggest based on a theoretical model that the distinctive elongated shape of a cilium may be coupled to its sensory function.

**Decision letter after peer review:**

Thank you for submitting your article "Ciliary chemosensitivity is enhanced by cilium geometry and motility" for consideration by *eLife*. Your article has been reviewed by 2 peer reviewers, and the evaluation has been overseen by a Reviewing Editor and Aleksandra Walczak as the Senior Editor. The reviewers have opted to remain anonymous.

The reviewers have discussed their reviews with one another, and the Reviewing Editor has drafted this to help you prepare a revised submission. We note that while the reviewers are generally supportive of the work, they have raised a number of pointed questions about the relevance of the results to biology. These include (1) the lack of consideration of heterogeneity in receptor distribution (there are many reasons why cilia are slender – not just for sensing), (2) the very large Peclet numbers required for not much enhancement in performance, and (3) a lack of biological relevance in the case of motile bundles.

*Reviewer #1 (Recommendations for the authors):*

The problem is interesting and the approach is valid. I have the following comments and questions about the work:

1. The main research question "why are so many chemical receptors located on cilia" (Line 228) seems ill-posed. Perhaps the question can be rephrased in terms of the effect of cilia geometry and motility on capture rate, which is tractable and quantifiable.

2. I appreciate that the authors quantify the amount of increase in capture rates by a cilium at rest and in shear flow compared to a surface patch. However, it is not clear to me whether in Figure 2(a) the authors use the classic analytical solution for the single cilium or their numerical solution. It is also not clear whether the classic analytical solution for the single cilium in a fluid at rest accounts for the presence of the reflecting wall. Please clarify and include a comparison between the analytical solution and the numerical solution in the presence of the wall.

3. The study of the effect of motility on the cilium capture rate could be developed further beyond the three examples shown in Figure 3. For example, how does the capture rate depend, not only on Pe, but also on the beating profile of the cilium. But my major concern with this study is that it considers ranges of Pe that are not achievable by the motion of a single cilium. I think a detailed discussion of the ranges of flow speeds generated by the beating cilium is in order here.

4. I have a similar comment on the case of collective pumping. The study considers only a single configuration with 7 cilia. Why 7? And Why this configuration? Further the range of Pe numbers considered also seems unrealistic for cilia-driven flows. What cilia-driven speeds are observed in the current model? How do they compare to cilia-driven flows reported in other experimental and computational studies. A comparison of the range of Pe to existing ranges of cilia-generated flow speeds would be very helpful here to gauge the validity of these large Pe.

5. I am particularly surprised by the author's claim that collective active pumping enhances capture rate. This is only true for extremely large Pe. The authors use Pe = 10000 to get 50% improvement in capture rate over cilia acting individually. This seems to suggest an unfavorable effect of hydrodynamic interactions among cilia for the purpose of chemical detection and seems to favor the hypothesis that solitary primary cilia are more suited for chemical detection.

6. Lines 252-253: the idea that randomly beating cilia are more suited for sensing and particle capture is already presented in Ding and Kanso 2015 and Nawroth et al. 2017 and should be acknowledged here. Note that these two references are cited by the authors but their contribution to the hypothesis that randomly-beating cilia enhance capture is not properly described. Only the fact that they considered finite sized particles is mentioned in lines 281 and 284.

7. The numerical algorithm uses a finite Reynolds number Re = 0.2, which is close but not consistent with cilia-driven flows. At the cilium level, Re is of the order 10^-6^ to 10^-4^.

8. Lines 272-273 and lines 276-277, the claim that Pe can exceed 10^4^ needs justification (see related comment above).

9. Paragraph 286-306 is awkward. It is not clear what is its purpose. If it is a motivational example, it is not clear why it is at the end of the manuscript. If the results in the manuscript are directly relevant to shed light on this system or suggest experiments (I don't think so), it should be stated explicitly.

10. Lines 307-309: I'm not sure if the series of examples considered in this manuscript constitute a proof that "the geometry of a cilium always means an advantage in chemical sensitivity over receptors on the epithelial surface, whether in a quiescent or moving fluid." This is true for the isolated cilium case but not straightforward to claim for interacting motile cilia.

11. In addition, it is my understanding that cilia also have receptors at their base. Please discuss.

*Reviewer #2 (Recommendations for the authors):*

1. In all cases considered by the authors, the capture rate of a cilium was comparable to a circular patch of larger (in some cases much larger) size, this cost-benefit analysis does not take into account the energetic costs of building/sculpting an elongated structure in the first place, compared to simply populating an area of flat membrane with receptors. Can the authors comment on this? Ideally with supporting calculations.

2. Many of the rate constants and other quantifiers show a strong length dependence, so one should expect to see such a dependence in experiments. The authors cite Challis et al. 2015 a few times in passing (line 117, line 157 etc) – a more detailed explanation and discussion of the Challis et al. experiments and findings could be useful here (type of cilia, length, function etc). Are there any other experimental studies that could support this, e.g. in other organisms?

3. Generally, the enhancement in capture rate for active cilia increases with Peclet number, but the degree of enhancement does not seem to be very large – so a high Pe is needed for a real effect. Please clarify the range of biologically relevant Peclet numbers? The discussions around this seemed to be very scattered. Pe = 10000 seems rather high (line 217), especially as most sensory (but motile) cilia operate in much more quiescent conditions and/or are involved in sensing very small molecules such as neuropeptides not large vesicles.

4. In the case of collective ciliary pumping, the authors found that the per-cilium improvement is higher when the cilia are not beating synchronously… how can this be reconciled with the fact that motile cilia in such a bundle will tend to synchronize by hydrodynamic interactions? Doesn't this mean that such bundles of motile cilia are therefore unlikely to be for useful chemosensing?

5. What about the distribution of receptors on a cilium? At the moment the authors assume and make comparisons with respect to uniform distributions only, related to the perfect absorber assumption – surely the elongated shape also presents opportunities for heterogeneous localisation of receptors. For instance, it is known that some receptors are restricted to the base of cilia, whereas other receptor types to ciliary tips.

6. Do cilia operate near the chemosensory limit? It is often said that detection of chemical molecules occurs at the physical limit in the motility apparatus of bacteria, what is the evidence that is happening with cilia? A more in-depth discussion with supporting numbers from the literature would be helpful.

---

## [Author Response]

Reviewer #1 (Recommendations for the authors):The problem is interesting and the approach is valid. I have the following comments and questions about the work:1. The main research question "why are so many chemical receptors located on cilia" (Line 228) seems ill-posed. Perhaps the question can be rephrased in terms of the effect of cilia geometry and motility on capture rate, which is tractable and quantifiable.

We agree that the question why something works the way it does is generally ill-posed in biology. We have reformulated the sentence as “does the location of so many chemical receptors on cilia bring them an advantage in sensitivity?”.

2. I appreciate that the authors quantify the amount of increase in capture rates by a cilium at rest and in shear flow compared to a surface patch. However, it is not clear to me whether in Figure 2(a) the authors use the classic analytical solution for the single cilium or their numerical solution.

Both numbers came from the analytical solution/approximation. However, based on the comment 7 (about the Reynolds number) we changed the analytical approximation. This also had some influence on the ratio, which is now 6. In the revised manuscript, we now state the source of the numbers clearly in the figure caption. We also compare the results against simulations (Figure 2, Suppl. 1).

It is also not clear whether the classic analytical solution for the single cilium in a fluid at rest accounts for the presence of the reflecting wall. Please clarify and include a comparison between the analytical solution and the numerical solution in the presence of the wall.

Yes, the analytical solution takes into account the reflecting boundary condition by expanding the space with a symmetric image. We now included the explanation: “First, we eliminate the reflective boundary condition at the surface by symmetrically extending the problem to a cylinder of length 2L in open space and considering 1/2 of its capacitance.”. A comparison with the static simulation result is now given in the sentence following Equation 4.

3. The study of the effect of motility on the cilium capture rate could be developed further beyond the three examples shown in Figure 3. For example, how does the capture rate depend, not only on Pe, but also on the beating profile of the cilium.

This is a very interesting proposition. We extended Figure 3 to include a beating pattern consisting of a straight working stroke and a sweeping recovery stroke (panel d). It performs very well at high Péclet numbers.

But my major concern with this study is that it considers ranges of Pe that are not achievable by the motion of a single cilium. I think a detailed discussion of the ranges of flow speeds generated by the beating cilium is in order here.

Cilia work in a range of different viscous and viscoelastic fluids. At the same time, the particles that are captured can range from small molecules to exosomes. We therefore expect a wide range of Péclet numbers, covering the full range of Péclet numbers discussed here. We have therefore included more example calculations into the manuscript.

4. I have a similar comment on the case of collective pumping. The study considers only a single configuration with 7 cilia. Why 7? And Why this configuration?

We considered 7 cilia arranged on a hexagon as a minimal example resembling a small ciliary bundle. The main message is that interaction with other cilia can enhance the capture rate on each of them. This finding is robust against details of the arrangement, as we now confirm by including other numbers (4 cilia on a square, 19 cilia on a hexagon; revised Figure 4 f,g), which all show the same trend.

Further the range of Pe numbers considered also seems unrealistic for cilia-driven flows. What cilia-driven speeds are observed in the current model? How do they compare to cilia-driven flows reported in other experimental and computational studies. A comparison of the range of Pe to existing ranges of cilia-generated flow speeds would be very helpful here to gauge the validity of these large Pe.

Examples of achievable Péclet numbers are now given in the discussion. They can be very large in viscous media and/or with larger particles. In the example with the (still hypothetical) proposed mechanism in Kupffer’s vesicle, we estimate Pe=1300, meaning that the capture rates could be significantly enhanced by the ciliary motility.

5. I am particularly surprised by the author's claim that collective active pumping enhances capture rate. This is only true for extremely large Pe. The authors use Pe = 10000 to get 50% improvement in capture rate over cilia acting individually. This seems to suggest an unfavorable effect of hydrodynamic interactions among cilia for the purpose of chemical detection and seems to favor the hypothesis that solitary primary cilia are more suited for chemical detection.

The answer to this question depends on whether the readout is the capture rate per cilium or the sum of capture rates by the whole bundle. Even if the capture rate is slightly reduced per cilium, the total capture rate is still higher for the bundle. We consider the finding that hydrodynamic interactions enhance the sensing of individual cilia as interesting in its own right, but the advantage of a bundle does not stand or fall by it.

6. Lines 252-253: the idea that randomly beating cilia are more suited for sensing and particle capture is already presented in Ding and Kanso 2015 and Nawroth et al. 2017 and should be acknowledged here. Note that these two references are cited by the authors but their contribution to the hypothesis that randomly-beating cilia enhance capture is not properly described. Only the fact that they considered finite sized particles is mentioned in lines 281 and 284.

We have now cited the finding from the references mentioned that randomly beating cilia have a higher capture rate in the Results section.

7. The numerical algorithm uses a finite Reynolds number Re = 0.2, which is close but not consistent with cilia-driven flows. At the cilium level, Re is of the order 10^-6^ to 10^-4^.

We realize that this is a misunderstanding. All our numerical calculations were carried out at zero Reynolds number, as indicated by the use of the Stokes equation and the Rotne-Prager approximation. For the very specific problem of particle capture on a moving cylinder, we used the assumption Re=0.2 as a way to avoid the problem of the Stokes paradox when making the cylinder infinitely long. Namely, the mobility of a laterally-moving cylinder diverges with its length at zero Reynolds number, but stays finite in inertial flows. Even though this workaround is a common solution, we have rewritten our calculation to avoid assuming a finite Reynolds number and changed the resulting values appropriately. We did this by using a force density on the cilium obtained from the well-established resistive force theory and then determine the capture rate on an infinite cylinder with the same force density, thus avoiding the problem of the Stokes paradox. Surprisingly, we could not find an exact solution to the problem of particle capture (i.e., calculation of the Nusselt number) on a cylinder with a given force density in the limit of high Péclet numbers, even though the problem is analytically solvable. We are now showing the complete derivation as Appendix 2.

8. Lines 272-273 and lines 276-277, the claim that Pe can exceed 10^4^ needs justification (see related comment above).

Examples of achievable Péclet numbers are now given in the discussion.

9. Paragraph 286-306 is awkward. It is not clear what is its purpose. If it is a motivational example, it is not clear why it is at the end of the manuscript. If the results in the manuscript are directly relevant to shed light on this system or suggest experiments (I don't think so), it should be stated explicitly.

We have greatly shortened the discussion of the left-right organizer and now use it as an example when discussing the Peclet numbers (which we estimate as 1300).

10. Lines 307-309: I'm not sure if the series of examples considered in this manuscript constitute a proof that "the geometry of a cilium always means an advantage in chemical sensitivity over receptors on the epithelial surface, whether in a quiescent or moving fluid." This is true for the isolated cilium case but not straightforward to claim for interacting motile cilia.

We agree that a comparison between sensing on a motile cilium and on a passive patch is not straightforward and is not made in this manuscript, although we do show that motility always leads to a sensitivity advantage. In the revised version the sentence states more precisely that we show the advantage of the cilium geometry over a patch for individual cilia.

11. In addition, it is my understanding that cilia also have receptors at their base. Please discuss.

The distribution of receptors along cilia could also be related to their trafficking and ectocytosis, and it is also not clear whether the target molecules bind directly to the receptor, or first non-specifically to the membrane. To our knowledge there is too little experimental evidence for any role of non-uniform receptor distribution.

From an esthetic point of view, Equations 3 and 5 seem unnecessary and this material could be moved to the methods section.

We agree with the Reviewer that the description of the electrostatic analogy interrupts the derivation. We have therefore moved all of it to a new Appendix 1.

Reviewer #2 (Recommendations for the authors):1. In all cases considered by the authors, the capture rate of a cilium was comparable to a circular patch of larger (in some cases much larger) size, this cost-benefit analysis does not take into account the energetic costs of building/sculpting an elongated structure in the first place, compared to simply populating an area of flat membrane with receptors. Can the authors comment on this? Ideally with supporting calculations.

We agree that the energetic cost of sensation and signal processing is a highly interesting topic. The surface patch with an equivalent capture rate often exceeds the available surface area. We therefore think that the relevant question is whether the increased sensitivity outweighs the cost, rather than the comparison between a cilium and a patch as such. Nevertheless, as our calculations primarily apply to higher organisms, the benefit is extremely difficult to estimate, and we also think that the cost of assembling and maintaining cilia, for example, for olfaction, is very small in the energy budget of the whole organism. We therefore think that any comparison would depend on too many uncertainties.

2. Many of the rate constants and other quantifiers show a strong length dependence, so one should expect to see such a dependence in experiments. The authors cite Challis et al. 2015 a few times in passing (line 117, line 157 etc) – a more detailed explanation and discussion of the Challis et al. experiments and findings could be useful here (type of cilia, length, function etc). Are there any other experimental studies that could support this, e.g. in other organisms?

In the revised manuscript, we expanded the discussion as suggested by the Reviewer. Unfortunately, we are not aware of any other publications on this question. We hope that our manuscript will provide some motivation for further studies.

3. Generally, the enhancement in capture rate for active cilia increases with Peclet number, but the degree of enhancement does not seem to be very large – so a high Pe is needed for a real effect. Please clarify the range of biologically relevant Peclet numbers? The discussions around this seemed to be very scattered. Pe = 10000 seems rather high (line 217), especially as most sensory (but motile) cilia operate in much more quiescent conditions and/or are involved in sensing very small molecules such as neuropeptides not large vesicles.

We now give examples of achievable Péclet numbers in the discussion.

4. In the case of collective ciliary pumping, the authors found that the per-cilium improvement is higher when the cilia are not beating synchronously… how can this be reconciled with the fact that motile cilia in such a bundle will tend to synchronize by hydrodynamic interactions? Doesn't this mean that such bundles of motile cilia are therefore unlikely to be for useful chemosensing?

Even when beating synchronously, a bundle of cilia reaches a somewhat higher capture rate per cilium than single beating cilium. The total rate is higher, in any case. Our main message is that the flow generated by other cilia enhances the capture rate on each of them, and that is the case in all scenarios we considered. The effect is stronger when the cilia beat out of phase, but it is present in other cases, too. Besides that, even if the capture rate per cilium only shows a slight increase, the bundle still has the advantage of containing many cilia. Coordination between cilia can have other advantages that are beyond our study.

5. What about the distribution of receptors on a cilium? At the moment the authors assume and make comparisons with respect to uniform distributions only, related to the perfect absorber assumption – surely the elongated shape also presents opportunities for heterogeneous localisation of receptors. For instance, it is known that some receptors are restricted to the base of cilia, whereas other receptor types to ciliary tips.

We now include numerical simulations that examine how localising sensors near the tip affects chemosensitivity. Specifically, we calculated the capture rate for cilia that are only absorbing on a fraction of 25%, 50% or 75% measured from the tip. The capture rates are naturally reduced, but less than the fraction of non-absorbing cilium length. In our opinion the localization of receptors is difficult to interpret, because we do not know if the target molecules are captured directly by the receptor, or do they bind non-specifically to the cilium membrane first.

6. Do cilia operate near the chemosensory limit? It is often said that detection of chemical molecules occurs at the physical limit in the motility apparatus of bacteria, what is the evidence that is happening with cilia? A more in-depth discussion with supporting numbers from the literature would be helpful.

The estimate in our discussion shows that the detection threshold for a single cilium without flow can be in the sub-picomolar range if the detection time is 1 second. The theoretical sensitivity further increases with the number of cilia and with flow, but decreases with medium viscosity and particle size. An exact quantitative comparison of sensitivities is difficult, but there are several studies that show sensitivities to concentrations (in the liquid phase) well below 1pM. We have now included two experimental references reporting single-neuron recordings with low thresholds: Frings and Lindemann (1990) and Zhang, Pacifico, Cawley, Feinstein and Bozza (2013).